# CAN WIKIPEDIA HELP OFFLINE REINFORCEMENT LEARNING?

## ABSTRACT

Fine-tuning reinforcement learning (RL) models has been challenging because of a lack of large scale off-the-shelf datasets as well as high variance in transferability among different environments. Recent work has looked at tackling offline RL from the perspective of sequence modeling with improved results as result of the introduction of the Transformer architecture. However, when the model is trained from scratch, it suffers from slow convergence speeds. In this paper, we look to take advantage of this formulation of reinforcement learning as sequence modeling and investigate the transferability of pre-trained sequence models on other domains (vision, language) when finetuned on offline RL tasks (control, games). To this end, we also propose techniques to improve transfer between these domains. Results show consistent performance gains in terms of both convergence speed and reward on a variety of environments, accelerating training by 3-6x and achieving state-of-the-art performance in a variety of tasks using Wikipedia-pretrained and GPT2 language models. We hope that this work not only brings light to the potentials of leveraging generic sequence modeling techniques and pre-trained models for RL, but also inspires future work on sharing knowledge between generative modeling tasks of completely different domains.

## 1 INTRODUCTION

Large pre-trained language models have shown impressive performance in natural language (Devlin et al., 2019; Radford et al., 2018) and vision (Dosovitskiy et al., 2021) tasks. Furthermore, Transformer-based autoregressive language models (Vaswani et al., 2017; Baevski & Auli, 2019; Radford et al., 2019) have shown to be powerful sources of zero-shot and few-shot performance (Brown et al., 2020), with notable rapid adaptation in low resource settings, demonstrating their easy adaptability and transferability to a number of tasks in their respective domains. Adapting autoregressive language models has also been extended to the multimodal setting (Tsimpoukelli et al., 2021) for tasks such as visual question answering.

Concurrently, offline reinforcement learning (RL) has been seen as analogous to sequence modeling (Chen et al., 2021; Janner et al., 2021; Furuta et al., 2021), framed as simply supervised learning to fit return-augmented trajectories in an offline dataset. This relaxation, doing away with many of the complexities commonly associated with reinforcement learning (Watkins & Dayan, 1992; Kakade, 2001), allows us to take advantage of techniques popularized in sequence modeling tasks for RL.

Pre-training, particularly, is an essential technique for alleviating higher compute costs from using more expressive models such as Transformers. However, such concept is still relatively fresh in RL (Singh et al., 2020; Tirumala et al., 2020), due to the difficulty in parameterizing different scenes and tasks through a single network (Wang et al., 2018b; Jiang et al., 2019; Zeng et al., 2020) as well as the lack of large off-the-shelf datasets for pre-training (Cobbe et al., 2020; Zhu et al., 2020; Yu et al., 2020). Adopting pre-training as a default option for recent Transformer-based methods (Chen et al., 2021; Janner et al., 2021; Furuta et al., 2021) appears far away – if we only look within RL.

Unified under the umbrella of sequence modeling, we look at whether Transformer-based pre-trained *language* models are able to be adapted to standard offline reinforcement learning tasks *that have no relations to language*. Given the setting of having a single model pre-trained on natural language to finetune on each offline RL task individually, we demonstrate drastic improvements in convergence speeds and final policy performances. We also consider further techniques (e.g. extension of positional

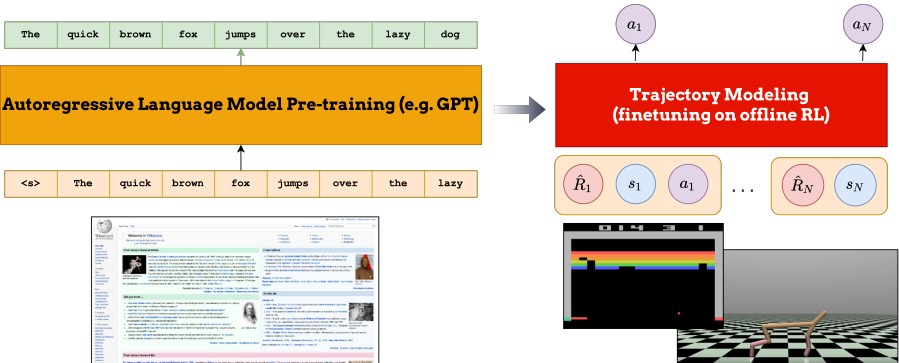

Figure 1: Adapting pre-trained language models (e.g. from Wikipedia) to offline RL (e.g. in continuous control and games).

embeddings, embedding similarity encouragement) in order to better take advantage of the features learned by the pre-trained language model and demonstrate greater improvements.

We demonstrate that pre-training on autoregressively modeling natural language provides consistent performance gains when compared to the Decision Transformer (Chen et al., 2021) on both the popular OpenAI Gym (Brockman et al., 2016) and Atari (Bellemare et al., 2013) offline RL benchmarks. We also note a significantly faster convergence speed, with a 3-6x improvement over a vanilla Decision Transformer turning hours of training to tens of minutes, indicating long-term computational efficiency benefits on language pre-training.

Our findings allude to the potential impact of large scale pre-training for reinforcement learning, given its surprising efficacy when transferring from a distant sequence modeling domain such as natural language. Notably, unlike other work on multi-task offline RL, our model provides consistent results in terms of both reward and convergence regardless of environment and setting, indicating a forseeable future where everyone should use a pre-trained language model for offline RL.

## 2 BACKGROUND

**Offline Reinforcement Learning**   We consider a standard Markov Decision Process (MDP) with state space $s \in \mathcal{S}$ and action space $a \in \mathcal{A}$, specified by a initial state distribution $p(s_1)$, a dynamics distribution $p(s_{t+1}|s_t, a_t)$, and a scalar reward function $r(s, a)$. The goal of reinforcement learning (RL) is to find the optimal policy $\pi^*(a|s)$ which maximizes the $\gamma$-discounted expected return as the agent interacts in the environment,

$$\max_{\pi} \mathbb{E}_{s_{1:\infty}, a_{1:\infty} \sim p, \pi} \left[ \sum_{t=1}^{\infty} \gamma^t r(s_t, a_t) \right] \tag{1}$$

In *offline* RL, the objective remains the same, but has to be optimized with no interactive data collection on a fixed set of trajectories $\tau_i$, each of the form below with horizon $N$,

$$\tau = (r_1, s_1, a_1, r_2, s_2, a_2, \dots, r_N, s_N, a_N). \tag{2}$$

Common approaches include value-based or model-based objectives with regularization (Fujimoto et al., 2019; Levine et al., 2020), and more recently, direct generative modeling of these trajectories conditioned on hindsight returns (Chen et al., 2021; Janner et al., 2021; Furuta et al., 2021).

**Transformer model**   In this subsection, we briefly review the Transformer architecture (Vaswani et al., 2017) used to model sequences. The Transformer is comprised of stacks of identical *Transformer layers*. Each of these layers takes in a set of $n$-dimensional vectors that are fed through the two main building blocks: a multi-head self-attention sublayer and a feedfoward MLP as shown below:

$$\text{Attention}(x) = \text{softmax}\left(\frac{Q(x)K(x)^{\top}}{\sqrt{n}}\right)V(x) \tag{3}$$

$$\text{Feedforward}(x) = L_2(g(L_1(x))) \tag{4}$$

where $Q, K$ and $V$ represent linear projections that parameterize the projection of input $x$ into the query, key and value spaces; while $L_1, L_2$ and $g$ represent the first linear projection, second linear projection, and activation function that comprise the feedforward MLP. This is followed by a residual connection (He et al., 2015) and layer normalization (Ba et al., 2016).

**Autoregressive Language Model Pre-training** Although there are now multiple techniques for language model pre-training (e.g. masked language modeling; Devlin et al., 2019), we will review autoregressive language modeling given its correspondence with the sequence modeling objective we employ for our offline reinforcement learning tasks.

Given a sequence $\mathbf{x} = [\mathbf{x}_1, \mathbf{x}_2, \ldots \mathbf{x}_N]$ comprised of tokens $\mathbf{x}_i$, we look to model the likelihood of the sequence $P(\mathbf{x})$ by way of modeling the probability of predicting each token $\mathbf{x}_i$ in a step-by-step or autoregressive fashion (commonly left-to-right). Naturally, it follows that each token's prediction will be conditioned on all the previous elements in the sequence $\mathbf{x}_{<i}$ as shown below (Bengio et al., 2001):

$$P(\mathbf{x}) = \prod_{i=1}^{N} p(\mathbf{x}_i | \mathbf{x}_{i-1}, \mathbf{x}_{i-2}, \ldots, \mathbf{x}_1) \tag{5}$$

## 3 METHODOLOGY

In this section we discuss our proposed methodology and techniques to better adapt pre-trained language models to model trajectories, as in the case of offline RL tasks with minimal modification to architecture and objectives shown in Figure 2.

### 3.1 MODELING

Following (Chen et al., 2021), we model trajectories autoregressively by representing them in the following manner:

$$\mathbf{t} = (\hat{R}_1, s_1, a_1, \hat{R}_2, s_2, a_2, \ldots, \hat{R}_N, s_N, a_N) \tag{6}$$

where trajectory $\mathbf{t}$ is modeled analogously to sequence $\mathbf{x}$ as shown in in Equation 5, and $\hat{R}_i = \sum_{t=i}^{N} r_t, s_i, a_i$ represent the returns-to-go, state and action for each timestep $i$ given $N$ timesteps, respectively.

### 3.2 TECHNIQUES

**Encouraging similarity between language representations and offline RL input representations**
We find the issue of lack of alignment between state, action and reward input representations and language representations — partially holding back further extraction of the capabilities of the language model. To this end, we use a similarity-based objective in order to maximize the similarity between the set of language embeddings $E = [E_1, \ldots, E_V]$ with vocabulary size $V$ and the set of input representations $I = I_1, \ldots, I_{3N}$. The input representations are parameterized by linear projections $L_r, L_a, L_s$ corresponding to the target reward projection, action projection and state projection, respectively.

Given the following cosine similarity function:

$$\mathcal{C}(z_1, z_2) = \frac{z_1}{\|z_1\|_2} \cdot \frac{z_2}{\|z_2\|_2} \tag{7}$$

we compute the negative (as we use gradient descent to optimize this objective) of the sum of the maximum similarity value for each embedding $E_1, \ldots, E_j, \ldots, E_V$ and each input representation $I_0, \ldots, I_i, \ldots, I_N$ as follows: [1]

$$\mathcal{L}_{\cos} = -\sum_{i=0}^{3N} \max_j \mathcal{C}(I_i, E_j) \tag{8}$$

---

[1]We looked at using mean pooling instead of max pooling for this objective and found that models with the mean pooling objective did not converge.

This allows us to encourage the input embeddings to become more similar to their language counterparts. However, due to computational cost of computing this loss for large values of $V$, we propose to use $K$-means clustering over the embeddings to reduce the size of $V$ to number of clusters $K$. We then treat the cluster centers akin to the original embeddings in order to compute our loss. Furthermore, we optimize this computation with vectorization.

**Language model co-training** We also experiment with continuing to train jointly on language modeling and trajectory modeling. This allows us to encouraging the model's transformer backbone to be able to handle both language and trajectories simultaneously. We refer to the standard negative log likelihood loss over each predicted token used for this objective as $\mathcal{L}_{LM}$.

## 3.3 FINAL OBJECTIVE

We now combine the objectives into the final objective $\mathcal{L} = \mathcal{L}_{\text{MSE}} + \lambda_1 \mathcal{L}_{\cos} + \lambda_2 \mathcal{L}_{\text{LM}}$. Where $\mathcal{L}_{\text{MSE}}$ represents the mean squared error loss (calculated between the predicted continuous actions, and the continuous actions contained in the dataset) used for the primary trajectory modeling objective (Chen et al., 2021), $\mathcal{L}_{\text{LM}}$ represents the negative log likelihood-based token prediction language modeling objective, and $\lambda_1, \lambda_2$ represent hyperparameters to control the weight of the cosine similarity loss and language modeling loss, respectively.

## 4 EXPERIMENTS

| Game | ChibiT | GPT2 | DT | CQL | QR-DQN | REM | BC |
|------|--------|------|-----|-----|--------|-----|-----|
| Breakout | $280.3 \pm 63.7$ | $\mathbf{287.8 \pm 78.5}$ | 267.5 | 211.1 | 21.1 | 32.1 | 138.9 |
| Qbert | $22.3 \pm 9.3$ | $22.5 \pm 12.8$ | 15.4 | $\mathbf{104.2}$ | 1.7 | 1.4 | 17.3 |
| Pong | $\mathbf{112.3 \pm 7.2}$ | $111.0 \pm 5.7$ | 106.1 | 111.9 | 20.0 | 39.1 | 85.2 |
| Seaquest | $2.9 \pm 0.3$ | $\mathbf{3.0 \pm 0.2}$ | 2.5 | 1.7 | 1.4 | 1.0 | 2.1 |

Table 1: Gamer-normalized scores for the 1% DQN-replay Atari dataset. We report the mean and variance across three seeds. Highest mean scores are highlighted in bold.

| Dataset | Environment | ChibiT | GPT2 | CLIP | iGPT | DT | CQL | TD3+BC | BRAC-v | AWR | BC |
|---------|-------------|--------|------|------|------|-----|-----|--------|--------|-----|-----|
| Medium Expert | HalfCheetah | $\mathbf{91.7 \pm 1.1}$ | $91.8 \pm 0.5$ | $91.3 \pm 0.4$ | $1.9 \pm 0.1$ | 86.8 | 62.4 | 90.7 | 41.9 | 52.7 | 59.9 |
| | Hopper | $110.0 \pm 1.2$ | $110.9 \pm 1.6$ | $110.2 \pm 0.1$ | $6.9 \pm 3.7$ | 107.6 | $111.0$ | 98.0 | 0.8 | 27.1 | 79.6 |
| | Walker | $108.4 \pm 0.2$ | $108.9 \pm 0.3$ | $108.5 \pm 0.6$ | $0.5 \pm 0.7$ | 108.1 | 98.7 | $110.1$ | 81.6 | 53.8 | 36.6 |
| Medium | HalfCheetah | $43.3 \pm 0.1$ | $42.8 \pm 0.1$ | $42.3 \pm 0.2$ | $1.5 \pm 0.1$ | 42.6 | 44.4 | $48.3$ | 46.3 | 37.4 | 43.1 |
| | Hopper | $82.1 \pm 4.6$ | $79.1 \pm 1.1$ | $66.9 \pm 0.9$ | $5.7 \pm 1.5$ | 67.6 | 58.0 | 59.3 | 31.1 | 35.9 | 63.9 |
| | Walker | $77.8 \pm 0.1$ | $78.3 \pm 1.5$ | $74.1 \pm 0.9$ | $0.4 \pm 0.4$ | 74.0 | 79.2 | $83.7$ | 81.1 | 17.4 | 77.3 |
| Medium Replay | HalfCheetah | $39.7 \pm 0.5$ | $40.3 \pm 2.3$ | $37.9 \pm 0.2$ | $1.6 \pm 0.1$ | 36.6 | 46.2 | 44.6 | $47.7$ | 40.3 | 4.3 |
| | Hopper | $81.3 \pm 5.0$ | $94.4 \pm 2.5$ | $85.8 \pm 0.3$ | $5.7 \pm 0.9$ | 82.7 | 48.6 | 60.9 | 0.6 | 28.4 | 27.6 |
| | Walker | $71.3 \pm 2.0$ | $72.7 \pm 1.2$ | $69.9 \pm 0.3$ | $9.1 \pm 7.7$ | 66.6 | 26.7 | $81.8$ | 0.9 | 15.5 | 36.9 |
| **Average (All Settings)** | | **78.3** | **80.1** | **76.3** | 3.7 | 74.7 | 63.9 | 75.3 | 36.9 | 34.3 | 46.4 |

Table 2: Results for D4RL datasets[3]. We report the mean and variance for three seeds. Language model pre-trainined models are consistently better than the Decision Transformer, and outperform/are competitive other baselines.

## 4.1 MODELS

**Pre-trained Models** We use the popular **GPT2**-small model to benchmark the impact of language-only pre-training. For direct comparison with the Decision Transformer (Chen et al., 2021), we also pre-train a language model with the same parameter count on the popular language modeling Wikitext-103 dataset (Merity et al., 2016), consisting of over 100 million tokens from full Wikipedia articles. We refer to this model as **ChibiT**.[4] Note that when we transfer a pre-trained model towards trajectory modeling on an offline RL dataset, we transfer all the Transformer layers and positional embeddings, while replacing the language token embeddings with the projections of the action, state and reward representations.

---

[4]"Chibi" means "small" or "mini" in Japanese.

To explore the effect of pre-training on vision datasets, we also study **CLIP** (Radford et al., 2021) and **ImageGPT** (Chen et al., 2020). CLIP is comprised of an image encoder and a text encoder, and trained to predict which caption matches with which image. While the text encoder is an autoregressive Transformer, the image encoder is a Vision Transformer, which is not autoregressive. Therefore, for the autoregressive setup of offline reinforcement learning, we use the pre-trained text encoder as our initializer, while discarding the image encoder part. ImageGPT is based on the same Transformer architecture as GPT2, but instead of language, it is trained on images unrolled into long sequences of pixels in an autoregressive manner.

**RL Baselines**    In addition to benchmarking our pre-trained language models, we compare to popular state-of-the-art offline RL algorithms as follows: Decision Transformer (DT) (Chen et al., 2021), CQL (Kumar et al., 2020), TD3+BC (Fujimoto & Gu, 2021), BRAC (Wu et al., 2019), and AWR baselines (Peng et al., 2019).

**Hyperparameters**    We use the following hyperparameters for our language model pre-training: the architecture is the same as that of (Chen et al., 2021) (128 model dim, 1 attention head, 3 layers), learning rate of 3e-4, a batch size 65536 tokens, for 6 hours (80000 steps), using a warmup schedule over the first 10000. We the same byte-pair encoding (BPE; Sennrich et al., 2016; Kudo & Richardson, 2018) as that used by GPT-2 (Radford et al., 2019). For our offline RL tasks, we follow the hyperparameters used by (Chen et al., 2021). For our additional objectives, we decay $\lambda_1, \lambda_2$, to reach $0.0$ each after 5000 steps. We tune initial values of $\lambda_1$ for values of $\{0.1, 0.2\}$ and $\lambda_2$ for values of $\{0.0, 0.2, 0.4\}$. We include additional details in the appendix.

We benchmark our models against the D4RL offline RL benchmark datasets (Fu et al., 2020) for the OpenAI Gym MuJoCo (Brockman et al., 2016) and Atari (Bellemare et al., 2013) tasks.

## 4.2    ATARI

We run our ChibiT and GPT2 models on the challenging Atari dataset (Bellemare et al., 2013). We use the four Atari tasks evaluated in (Agarwal et al., 2020), namely Breakout, Qbert, Pong and Seaquest. Baseline numbers used are provided by (Chen et al., 2021) for behavior cloning and Decision Transformer models, while CQL, REM, and QR-QDN baseline numbers are provided by (Kumar et al., 2020; Agarwal et al., 2020). Following (Hafner et al., 2021), we normalize scores based on that of a professional gamer on the evaluation set.

We show results in Table 1. It can be seen that ChibiT and GPT2 results consistently improve over/match a strong vanilla Decision Transformer baseline. Our models are competitive with the Decision Transformer on all four games and competitive with CQL on 3/4 games.

## 4.3    GYM

In this section, we consider results on the OpenAI Gym tasks (HalfCheetah, Walker2d, and Hopper) from the D4RL benchmark (Fu et al., 2020).

We train our models for a total of 100k timesteps and evaluate every 5000 timesteps, with each evaluation consisting of 10 episodes. Baseline results are obtained directly from the D4RL paper (Fu et al., 2020) and Decision Transformer results are directly taken from (Chen et al., 2021). Similarly, following (Fu et al., 2020), we compute the normalized score over returns, computed by taking $100 \times \frac{\text{score} - \text{random score}}{\text{expert score} - \text{random score}}$.

We show results comparing ChibiT, GPT2, and CLIP with state-of-the-art offline RL algorithms in Table 2. Pre-training improves the Decision Transformer by large margins in an overwhelming majority of tasks, clearly demonstrating that language pre-training improves over random initialization using sequence modeling techniques in terms of reward. We also take note of the minimal difference between ChibiT, CLIP, and GPT2, showing that that at this scale, improvements on offline RL are not necessarily strongly correlated with model size as has been shown on both large-scale vision and language tasks. We note that CLIP, while improving over a vanilla DT model, is often slightly less competitive that our pure language modeling objectives. Our ChibiT and GPT2 models achieve and average performance of $78.3$ and $80.1$, respectively, showing strong competitiveness on all settings with all baselines. These pre-trained language models acheive state-of-the-art results by

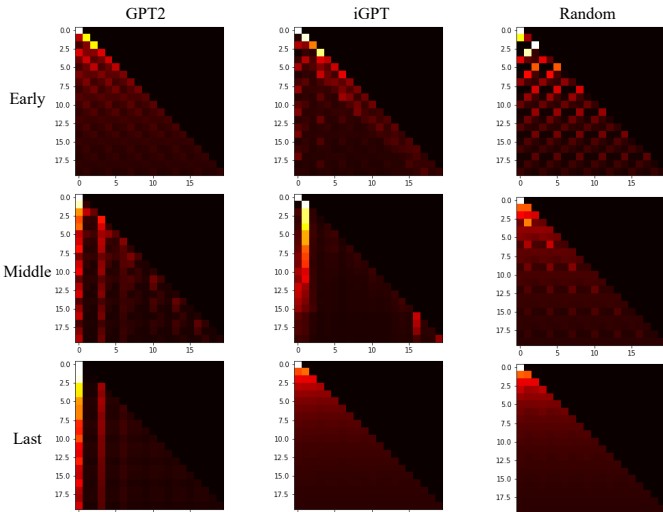

Figure 2: **Attention analysis.** We visualize early, middle and last attention weights computed by GPT-2, iGPT, and randomly initialized DT models on Hopper-medium to study how pre-training on different modalities affects how the model attends to previous timesteps. The x-axis represents keys (representations that are being "looked at") while the y-axis represents queries (i.e. representations that are "looking at" other representations) for a given timestep. Ligher colors represent higher attention weights, while darker colors represent lower weights.

outperforming the strong Decision Transformer and TD3+BC baselines by a significant 3.0-5.4 points.

## 5 ANALYSIS

In this section, we look at more fine-grained details and properties of various aspects of adapting pre-trained language models to offline RL tasks with ablations on OpenAI Gym.

**Convergence Speed**   We evaluate time-to-convergence of GPT2, ChibiT and DT using the our implementations of the former two and the author-provided implementation of the latter.

Results are reported in Table 3. We find that pre-training on language allows us to speed up the training process of Transformer-based offline RL models, measured in wall-clock time. Convergence is defined as the point where average performance attains a score within 2 (normalized score) of the best score.

Interestingly, we also find that GPT2, despite its larger model size at 84M model parameters, still manages to train faster than DT. This points towards potential benefits of pre-training at scale and increased efficiency during finetuning. We run experiments on a single NVIDIA V100 16GB GPU and an Intel Xeon Gold 6148 Processor.

**Language initialization versus vision initialization**
As we establish that Transformers pre-trained on language data are surprisingly effective for accelerating training convergence time on offline reinforcement learning tasks, it is tempting to ask if this phenomenon is inherent to language pre-training or does it extend to vision pre-training as well. To answer this question, we compare two GPT models, ImageGPT-small (iGPT) and GPT2-small (GPT2), pre-trained on language and vision data, respectively. Since Transformer architectures are domain-agnostic, these models can be trained on 1D sequences of any form. Hence, we can compare GPT2, which was pre-trained on many

| Model | Walker2d | HalfCheetah | Hopper |
|---|---|---|---|
| DT (GitHub) | 3h14m | 3h23m | 2h47m |
| ChibiT (ours) | 43m | 48m | 36m |
| GPT2 (ours) | 1h27m | 1h32m | 1h2m |

Table 3: Training time comparison (measured in hours and minutes on a single V100 GPU on the medium-expert setting) between the Decision Transformer and two pre-trained models: ChibiT and GPT2 on OpenAI gym tasks. Note that GPT2 is 144x larger than the other models with 84M model parameters.

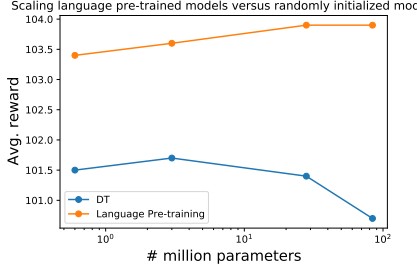

| Model | Avg. Reward |
|---|---|
| ChibiT (context = 20) | 67.7 |
| ChibiT (context = 60) | 67.3 |
| DT (context = 20) | 61.4 |
| DT (context = 60) | 61.2 |

Figure 3: Comparison of Average *Medium-Expert* reward for various model sizes on OpenAI Gym.

Table 4: Experiment on increased context length with pre-trained models on the medium setting

sequences of discrete language tokens, and iGPT, which was pre-trained on autoregressive image generation at the pixel level (note that both models were trained on $\sim 10^{10}$ tokens). Given the results in Table 2 for iGPT, we found that the model had extremely low returns, and did not reach convergence. Notably, on some seeds, the model even performed worse than a random score after training on Walker medium, with a normalized score of $-0.1$, in contrast with GPT-2 pre-training which gives us an average increase of $5.1$ points (measured in terms of normalized reward) over the Decision Transformer.

Furthermore, when we turn our attention to the difference between GPT2 and CLIP, we see that GPT2, which is based on pure-language based pre-training, performs better. While the text encoder of CLIP is also an autoregressive Transformer pre-trained on text, the objective of CLIP is different from GPT2 in that the former attempts to match text with a corresponding image, while the latter is pre-trained on pure autoregressive language modeling. Given this, we hypothesize that generative (versus discriminative) training objective is more useful for transfer to a generative task.

We believe that this alludes to underlying similarities between language modeling and trajectory modeling, whereas a large difference between image modeling and trajectory modeling. Perhaps this can be attributed to the "natural" sequential nature of language and trajectories, versus the forced 2D→1D nature that was used to pre-train iGPT.

**Attention Analysis**   To further understand the discrepancy between language-based and vision-based pre-training, we visualize attention weights, extracted from GPT2 and iGPT after fine-tuning on Hopper medium, as an example offline RL task. As a reference, we also extract attention weights from randomly initialized networks of Decision Transformers. In Figure 4.2, we plot the attention weights averaged over all attention heads in each model, and present the visualizations for early, middle, and last layers, respectively. Due to the autoregressive nature of our task, attention weights in the upper right triangle are masked out, so that the model can only attend to past sequences.

As a general trend, we see that in earlier layers GPT2 and the randomly initialized model tend to attend to positions with multiples of 3 timesteps behind the current position. This indicates that actions attend to previous actions, states attend to previous states, and returns-to-go attend to previous returns-to-go. Constrasted with this, iGPT's attention is less interpretable, however showing a notably stronger recency bias. In the middle layers, DT continues the trends of its early layers, whereas iGPT tends to fixate on a single state (given the overwhelming brightness of timestep 2), GPT2 starts showing a stronger preference for previous returns to go (given that lighter colors are consistently timestep 1, 4, etc...). Finally, in the models' last layer, while iGPT and random initialization tend to exhibit a behaviour closer to mean pooling over all previous inputs, GPT's final prediction seems to be heavily reliant on the initial returns-to-go. This perhaps indicates that goal conditioning is stronger in GPT2.

**How important is the model size of Transformer?**   We explore how pre-training changes the impact on model size for these offline RL tasks. We train randomly initialized models with various parameter counts (approx. 600K, 3M, 18M, 84M) as well as language-pre-trained models on WikiText-103 with the same parameter counts. Exact hyperparameters for this experiment are given in the Appendix.[5]

---

[5]Note that when pre-training language models with 600K, 3M, and 18M parameters, we control that our pre-training takes exactly 6 hours on 4 V100 GPUs.

| Model | HalfCheetah | Walker2d | Hopper |
|---|---|---|---|
| ChibiT (FT) | $43.3 \pm 0.1$ | $77.8 \pm 0.1$ | $82.1 \pm 4.6$ |
| ChibiT (Frozen) | $26.4 \pm 1.2$ | $63.3 \pm 2.7$ | $57.7 \pm 7.0$ |

Table 5: Experiment on freezing model weights versus finetuning them on OpenAI Gym.

| Model | HalfCheetah | Walker2d | Hopper |
|---|---|---|---|
| ChibiT | $43.3 \pm 0.1$ | $77.8 \pm 0.1$ | $82.1 \pm 4.6$ |
| ChibiT (w/o $\mathcal{L}_{\cos}$) | $43.1 \pm 0.1$ | $77.2 \pm 1.3$ | $80.9 \pm 1.1$ |
| ChibiT (w/o $\mathcal{L}_{\text{LM}}$) | $43.3 \pm 0.2$ | $77.6 \pm 0.2$ | $81.4 \pm 5.2$ |
| ChibiT (rand. pos. emb.) | $43.0 \pm 0.4$ | $76.5 \pm 1.2$ | $78.4 \pm 2.0$ |

Table 6: Ablation of our proposed techniques

We visualize the average (over Hopper, Walker2d, and HalfCheetah) of Medium-Expert results in Figure 3. Unsurprisingly, we observe that a randomly initialized Decision Transformer, tends to have lower relative returns as parameter sizes increase likely due to overfitting on finite data. Interestingly, however, pre-trained language models tend to increase performance as parameter count increases, despite diminishing returns with increasing parameter count. Nonetheless, this is exciting as it demonstrates that even language pre-training may be beneficial at scale, especially for larger and more diverse offline RL datasets in the future.

**Context length** We try various context lengths with pre-training and not pre-training: context = 20 (following (Chen et al., 2021)) and context = 60. Results are shown in Table 4. It can be seen that additional context does not seem to help even when pre-training on long range language modeling, perhaps alluding to the limited utility of long-range context for the OpenAI Gym tasks.

**Can we freeze model parameters?** We also look at how ChibiT performs when model weights (transformer blocks: self-attention and feedforward) are frozen with only action, state and return projections $L_a, L_s, L_r$ being trained. Previous work (Tsimpoukelli et al., 2021; Lu et al., 2021) has demonstrated how frozen language models have the capability to extend to the vision domain with respectable performance, which we aim to test with this experiment. We show results on Table 5 on the D4RL medium setting in OpenAI Gym. When freezing model weights, performance is underwhelming with performance drastically reducing as much as $\sim$40%. We conjecture this is due to our tasks being complex generative modeling as opposed to discriminative classification (Lu et al., 2021), where the output distribution is of a higher dimension — hence the need for more intensive finetuning.

**Ablation of proposed techniques** We perform an ablation study of our proposed auxiliary techniques and compare the impact of including and not including pre-trained positional embeddings. Results are shown in Table 6. It can be seen that the combination of our objectives are able to increase performance consistently. We also note that the removal of pre-trained positional embeddings results in the largest average decrease in performance over ChibiT, alluding to the fact that this positional information is important and transferable to offline RL.

## 6 RELATED WORK

**Transformer Pre-training** Pre-training Transformer-based models (Vaswani et al., 2017) was initially proposed by (Radford et al., 2018) with their Generative Pre-trained Transformer (GPT). They performed autoregressive language modeling on a relatively large dataset, showing promising initial success not only on its ability to scale to large models sizes, but also for its impressive performance when fine-tuning on task-specific natural language understanding (NLU; Wang et al., 2018a) datasets. BERT (Devlin et al., 2019), extended this pre-train→finetune paradigm with their masked language modeling objective. Furthermore, recently this paradigm has extended to computer vision with the Vision Transformer (ViT; Dosovitskiy et al., 2021) and iGPT (Chen et al., 2020) .

**Sequence Modeling for Offline RL** Offline RL became popular starting from a simple observation that many performant off-policy algorithms (Mnih et al., 2015; Lillicrap et al., 2015; Gu et al., 2016; Haarnoja et al., 2018; Fujimoto et al., 2018) fail to learn in a fully off-policy, i.e. *offline*, batch setting (Fujimoto et al., 2019). Numerous algorithmic work ensued (Wu et al., 2019; Jaques et al., 2020; Ghasemipour et al., 2021; Kumar et al., 2020; Fujimoto & Gu, 2021) with various applications (Jaques et al., 2020; Chebotar et al., 2021). Building on reward-conditioned imitation learning (Srivastava et al., 2019; Kumar et al., 2019), Transformers (Parisotto et al., 2020) have been

recently adopted for replacing offline RL with sequence modeling (Chen et al., 2021; Janner et al., 2021; Furuta et al., 2021). Despite initial successes, many techniques popular in language modeling have yet to be experimented in these offline RL benchmarks, and our work constitutes an initial step toward bridging the two communities.

**Pre-training for RL**   Contrary to language or vision (Devlin et al., 2019; Dosovitskiy et al., 2021), successes in deep RL have largely focused on isolated tasks/ domains (Mnih et al., 2015; Silver et al., 2016; Gu et al., 2017; Kalashnikov et al., 2018; Vinyals et al., 2019). Pre-training results are often limited to vision or language processing (Yen-Chen et al., 2020; Lynch & Sermanet, 2021) or specially-crafted domains (Singh et al., 2020; Tirumala et al., 2020). Arguably, a fundamental bottleneck for pre-training in RL is the difficulty in reusing a single network across vastly different tasks, observation spaces, action spaces, rewards, scenes, and agent morphologies. Preliminary work explored various aspects of this problem through graph neural networks for morphology generalization (Wang et al., 2018b; Pathak et al., 2019; Chen et al., 2018; Kurin et al., 2020), language for universal reward specification (Jiang et al., 2019; Lynch & Sermanet, 2021; Shridhar et al., 2022), and object-centric action spaces (Zeng et al., 2020; Shridhar et al., 2022; Noguchi et al., 2021). Our work is orthogonal to these as we essentially amortize RL algorithm itself, expressed as sequence modeling with Transformer, instead of specific RL domain information, and can be combined with domain-specific pre-training techniques (Yen-Chen et al., 2020; Lynch & Sermanet, 2021; Banino et al., 2021) effortlessly.

**Adapting language models to new modalities and domains**   Within language modeling recently there has been interest in adaptation of pre-trained language models by way of continued pre-training (Gururangan et al., 2020). Furthermore, (Tsimpoukelli et al., 2021) looked at adapting frozen language models for few-shot question answering by adding an auxiliary vision encoder. Other (concurrrent) work has proposed using language as a semantically meaningful way of communicating between modalities directly using frozen pre-trained language models for planning (Zeng et al., 2022; Li et al., 2022; Huang et al., 2022). More related to our work is that of (Lu et al., 2021), where they look at adapting frozen language models to various tasks such as image classification. Concurrent work (Reed et al., 2022) has looked at multi-tasking using generic sequence modeling for transformer-based RL agents, while other concurrent work has shown that language pre-training is helpful for in-context learning as a result of having a long-tailed distribution (Chan et al., 2022). Our work extends on the spirit of these works by adapting language models to a new domain of RL, however, as far was we know, we are the first to propose leveraging a generative model (in language) for generation in another domain (RL) as opposed to a discriminatory task such as classification.

## 7   CONCLUSION

We investigate how pre-trained models can improve generic offline RL problems, recently casted as sequence modeling. To our surprise, we discover that fine-tuning from a Wikipedia-trained small transformer (ChibiT) or a GPT2 model outperforms the basic Decision Transformer (DT) and other RL-based offline baselines by a large margin in terms of policy performance and convergence, establishing state-of-the-art scores on the competitive D4RL benchmark in both Gym and Atari and cutting down the DT training time by 3-6x. We perform extensive ablation studies and analyses, and found how language pre-training (as opposed to vision pre-training), model size, and fine-tuning (as opposed to freezing parameters) play critical roles in the final performances. We hope our work can accelerate the adoption of pre-training in RL and leads to more interest in applying other sequence modeling techniques from language and vision into RL.

Beyond RL, our work constitutes the first successful transfer, to the best of our knowledge, of a pre-trained generative model in one domain (language) to a generative modeling task in a completely different domain (RL on continuous control and games). This hints at some underlying universal structure across sequence modeling domains, and could perhaps lead to unified generative modeling pre-training for better transferability among them. In future work, we look to investigate in more depth which properties of language structure are useful for reinforcement learning and sequence modeling in other domains, and whether previous work studying language structure (Hupkes et al., 2019) does indeed relate to compositional generalization of neural networks.

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

# A APPENDIX

## A.1 HYPERPARAMETERS & TRAINING DETAILS

| Hyperparameter | Value |
|---|---|
| # Layers | 3 |
| # Attention Heads | 1 |
| Activation fn. | ReLU |
| Batch size | 64 |
| Context | 20 |
| Return-to-go conditioning | 6000 HalfCheetah |
| | 3600 Hopper |
| | 5000 Walker |
| Dropout | 0.2 |
| Learning rate | 1e-4 |
| LR Warmup | 5000 steps |
| $K$ for GPT2 | 500 |
| $K$ for ChibiT | 1000 |
| $\lambda_1$ | 0.1 |
| $\lambda_2$ | 0.2 |
| Hopper $\lambda_1$ | 0.2 |

Table 7: Hyperparameters used for OpenAI Gym

**On choosing the value of $K$**  We base the choice of the value of $K$ based on GPU memory constraints. For $K = 1000$ and $K = 500$, we find that they perform similarly in practice (both time and performance wise), albeit $K = 1000$ performing slightly better performance wise. However the memory requirements of $K = 1000$ tend to double – which often leads to OOM errors on a our NVIDIA V100 16GB GPUs for GPT-2 (motivating our reason to use $K = 500$ for GPT-2 and $K = 1000$ for ChibiT).

**Other implementation details**  Pre-trained models are trained with and taken from the HuggingFace Transformers library (Wolf et al., 2020). The model code for our GPT2 model is `gpt2`, CLIP is `openai/clip-vit-base-patch32`, and iGPT `openai/imagegpt-small`.

| Model | Parameter Count | Num. Tokens |
|---|---|---|
| DT | 596K | — |
| ChibiT | 596K | $10^7$ |
| iGPT | 84M | $10^{10}$ |
| GPT-2 | 84M | $10^{10}$ |
| CLIP | 38M | $10^{10}$ |

Table 8: Model parameter counts and number of unique pre-training tokens

**Language Model Pre-training with larger sizes**  For our large sized pre-trained models in our model scale experiments, we use the following dimensions:

| Param. Count | Model Dim. | Num. Heads | Num. Layers |
|---|---|---|---|
| 3M | 256 | 4 | 4 |
| 18M | 512 | 8 | 6 |
| 84M | 768 | 12 | 12 |

Table 9: Parameter count for various pre-trained models used in our model scale experiments.

## B  ATTENTION VISUALIZATION

We visualize the attention weights with a temperature of $0.1$ to improve visual interpretation.

## C  REPRODUCTION OF DT RESULTS VERSUS DT RESULTS IN (CHEN ET AL., 2021)

We re-run the results in (Chen et al., 2021) and include them for reference in Table 10.

| Dataset | Environment | DT | DT(ours) |
|---|---|---|---|
| Medium Expert | HalfCheetah | $86.8 \pm 1.3$ | $86.5 \pm 0.8$ |
| | Hopper | $107.6 \pm 1.8$ | $107.4 \pm 2.0$ |
| | Walker | $108.1 \pm 0.2$ | $108.4 \pm 0.1$ |
| Medium | HalfCheetah | $42.6 \pm 0.1$ | $42.1 \pm 0.3$ |
| | Hopper | $67.6 \pm 1.0$ | $68.1 \pm 3.1$ |
| | Walker | $74.0 \pm 1.4$ | $74.4 \pm 1.9$ |
| Medium Replay | HalfCheetah | $36.6 \pm 0.8$ | $36.2 \pm 1.4$ |
| | Hopper | $82.7 \pm 7.0$ | $80.4 \pm 6.3$ |
| | Walker | $66.6 \pm 3.0$ | $67.0 \pm 2.4$ |
| **Average (All Settings)** | | 74.7 | 74.5 |

Table 10: Re-implementation of Decision Transformer using their codebase[a]

[a]https://github.com/kzl/decision-transformer

## D  PERFORMANCE PROFILES

We compute statistical significance tests using `rliable` (Agarwal et al., 2021) on OpenAI Gym. Specifically, as we are only comparing two algorithms DT (Chen et al., 2021) and ChibiT, we only plot performance profiles and the boostrapped confidence interval measure.

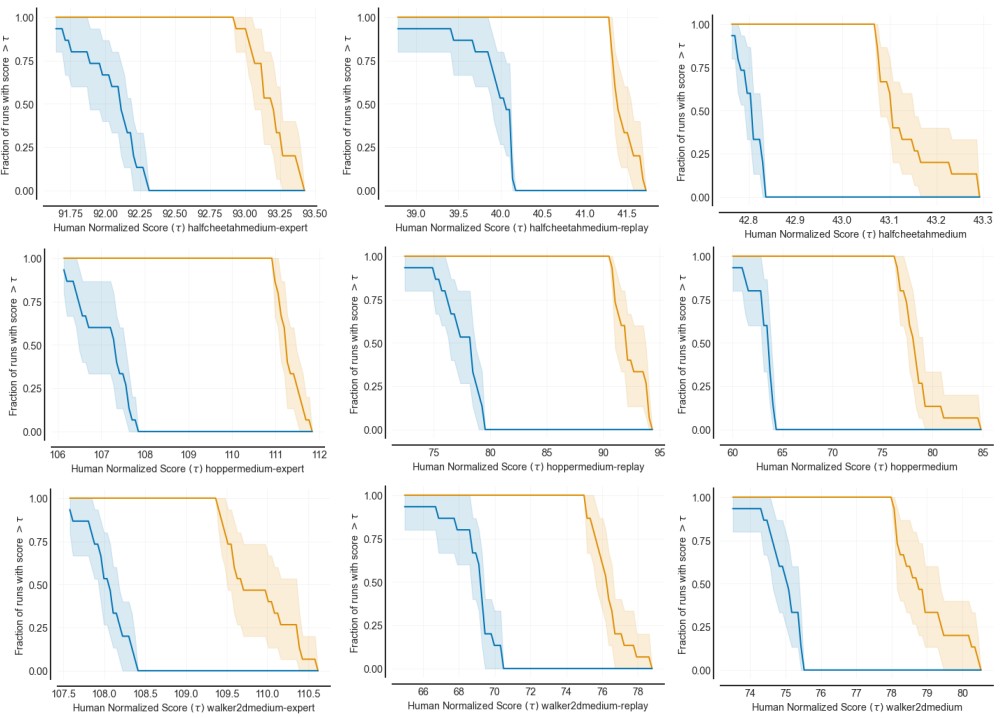

Figure 4: **Performance profiles on D4RL datasets.** Yellow colors represent ChibiT and blue colors represent Decision Transformer (DT). We report the profiles based on score distributions over 10 runs using different random seeds. Language model pre-trained models are consistently better than DT.

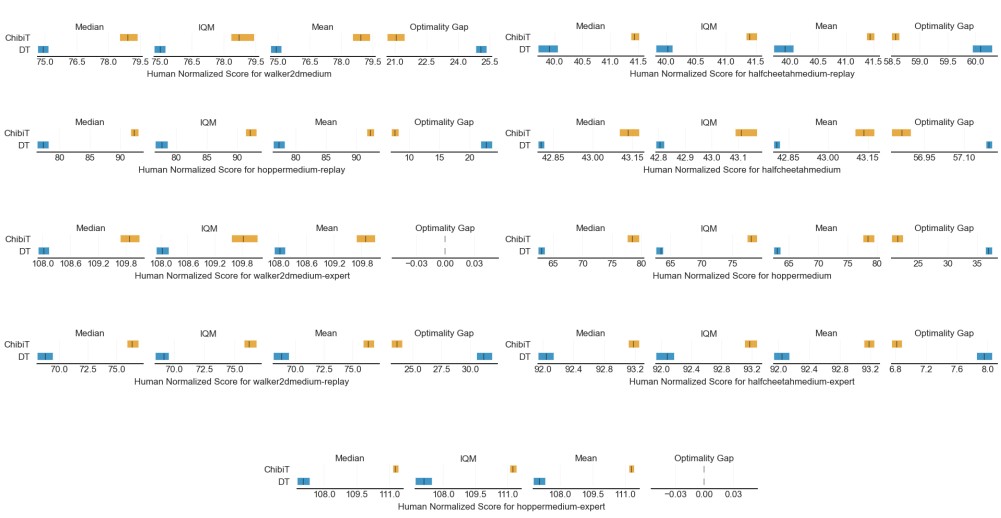

Figure 5: Bootstrapped confidence intervals (CIs) on D4RL datasets. Yellow colors represent ChibiT and blue colors represent Decision Transformer (DT). We report the intervals based on score distributions over 10 runs using different random seeds. Language model pre-trained models are consistently better than DT.

