# OpenReview forum: "Can Wikipedia Help Offline Reinforcement Learning?"
_ICLR.cc/2023/Conference — Submitted to ICLR 2023_

### Official Review · Reviewer_3DmF · 2022-10-25

**Confidence:** 4
**Correctness:** 3
**Technical Novelty And Significance:** 4
**Empirical Novelty And Significance:** 3
**Recommendation:** 6

**Clarity, Quality, Novelty And Reproducibility:**

Well written paper with no reproducibility concerns and thorough empirical evaluation.  The results presented are novel and broadly relevant.

**Strength And Weaknesses:**

Strengths:
- Provides strong evidence that pretraining on language at least matches but more often improves performance over random initialization, often by a large margin.
- Good experimental design and thoroughness, specifically: Training ChibiT to give a comparably sized model to DT which enabled apples to apples comparison and an understanding of the impact of model size on transferability, reproducing DT results to ensure comparability, using rliable for statistical significance tests, including the attention analysis & thorough ablations

Weaknesses:
- Not much motivation is given for using the LM objective. I find it surprising that this improves performance, do you have intuition for why this is?  I see in Table 6 that removing either the LM objective or the representation alignment objective hurts performance, and I wonder at the impact of removing both.  A hypothesis could be that these two objectives work in conjunction, that pushing the RL representations to align with the original word embeddings makes it beneficial for the transformer to continue to be able to act on the original word embeddings.  If this is the case then removing one or the other alone would clearly harm performance, but it wouldn’t necessarily be the case that removing both would harm performance.

Small notes & questions (that don’t impact decision):
- In Section 4.1, Hyperparameters paragraph, sentence 2 is missing a word: “We [use] the same byte-pair encoding…”
- In Section 5, “Can we freeze model parameters?” You conjecture that freezing hurt performance in your setting in contrast to Lu et al 2021 because your task is more complex (a plausible conjecture).  However, note that a follow-up to the Lu et al 2021 paper (Arxiv: https://arxiv.org/abs/2107.12460) indicates that when freezing model weights, performance reduces on the subset of tasks from Lu et al 2021 tested (if the learning rate is tuned). This suggests that a difference in task complexity may not be the sole reason for freezing hurting performance in your case.


**Summary Of The Paper:**

This paper evaluates the impact of using language and vision pretrained transformers as an initialization for a Decision Transformer style model used in offline RL settings.  It provide methods for improving the transferability of the pretrained models and demonstrate that pretraining consistently matches or outperforms random initialization on Atari and substantially outperforms random initialization on D4RL, in addition to accelerating training speed.  It also includes thorough ablations demonstrating the value of the proposed methods and additional insight into the impact of design decisions like pretraining modality and model size among others.

**Summary Of The Review:**

A nice paper that demonstrates the effectiveness of language model pretraining for offline RL applications by proposing methods to facilitate this transfer and providing thorough and compelling experiments supporting the success of their approach.

---

> ### Author Response · Authors · 2022-11-18
> **Response**
>
> We are happy that the reviewer finds our paper to have strong evidence for our hypothesis and thorough evaluation. We look to address questions below:
>
>
> > Not much motivation is given for using the LM objective. I find it surprising that this improves performance, do you have intuition for why this is? I see in Table 6 that removing either the LM objective or the representation alignment objective hurts performance, and I wonder at the impact of removing both. A hypothesis could be that these two objectives work in conjunction, that pushing the RL representations to align with the original word embeddings makes it beneficial for the transformer to continue to be able to act on the original word embeddings. If this is the case then removing one or the other alone would clearly harm performance, but it wouldn’t necessarily be the case that removing both would harm performance.
>
> We appreciate a useful insight regarding the LM and alignment objectives. However, we note that ChibiT (rand. pos. emb.) in Table 6 refers to the case where we remove both LM and alignment objectives, but we see the lower performance. We do agree that the two objectives work in conjunction to help improve the model.
>
>
> > In Section 5, “Can we freeze model parameters?” You conjecture that freezing hurt performance in your setting in contrast to Lu et al 2021 because your task is more complex (a plausible conjecture). However, note that a follow-up to the Lu et al 2021 paper (Arxiv: https://arxiv.org/abs/2107.12460) indicates that when freezing model weights, performance reduces on the subset of tasks from Lu et al 2021 tested (if the learning rate is tuned). This suggests that a difference in task complexity may not be the sole reason for freezing hurting performance in your case.
>
> Thank you for pointing this out. Though learning rate may impact performance to some extent, we note the margin of freezing versus not freezing is quite large and believe that learning rate tuning would not make a significant difference.

---

> > ### Comment · Reviewer_3DmF · 2022-11-27
> > **Reviewer Response**
> >
> > Thank you for your responses
> > - Regarding the lack of motivation for using the LM objective: See below for a more extensive response, but even beyond the question of effectiveness of the technique, I believe that a descriptive explanation for the motivation behind choosing the additional language modeling objective is still missing from the paper.
> > - Regarding the effect of freezing: I agree with you. My initial comment was poorly phrased, I meant "From reading prior work I would expect freezing to hurt performance, so I don't find your result surprising and don't believe that you need to justify it."  Regardless, this was mainly a (possibly unnecessary) aside.
> >
> > Overall however, after considered your response to the motivation around the LM objective and closely considering the other reviewers concerns, I believe I was initially mis-calibrated in my initial score and therefore am lowering it because while I believe this is a novel contribution that partially answers the question "can english pretraining improve performance on RL tasks?", I have two significant remaining concerns:
> > 1) You make multiple claims as if they hold generally, but they are only investigated wrt D4RL because the Atari results indicate that using language pretraining is only competitive with training from random initialization.  The lack of Atari results (on, for example, convergence speed) means that this paper is predominantly about exploring the effect of language pretraining on D4RL which limits the generality of conclusions we can draw from it.
> >  2) Digging into your details about experimental procedure and the ablations around the additional objectives, I believe that they have a very minor impact on the claims of the paper and that this isn't clear in the paper presentation.  Because these are presented as the main technical contributions of the paper, it is a bit misleading that they contribute very little to the improved performance.
> >       - From the experimental perspective: If you decay the lambdas to 0 after 5k steps then you are only using them for the first 5% of training which isn't clear from the presentation in the methodology section.  The statement in 3.2 Language Model Co-Training of "This allows us to encouraging the model’s transformer backbone to be able to handle both language and trajectories simultaneously" may be technically true but this doesn't describe the target behavior at the end of training, only the goal during the first half of your warmup period.
> >       - From the ablation perspective: Considering the results in Table 6, from your comment my understanding is that the 4th line ("ChibiT rand. pos. emb.") is the baseline without any of your objectives.  Then, using both of the objectives (the top line) is only statistically significantly better than this baseline in 1 out of 3 cases (Walker2d).  Additionally, removing just the LM objective is never statistically significantly worse than using both objectives (line 1 vs line 3). But more importantly, the small gains in mean score, significant or not, are very small in comparison to the amount that ChibiT outperforms Decision Transformer in nearly all cases.

---

### Official Review · Reviewer_y7eL · 2022-10-25

**Confidence:** 3
**Correctness:** 3
**Technical Novelty And Significance:** 2
**Empirical Novelty And Significance:** 3
**Recommendation:** 6

**Clarity, Quality, Novelty And Reproducibility:**

# Clarity

The paper is well-written and easy to follow

# Quality and Novelty

Inspired by the recent success of reformulating RL as a sequence modeling problem, the investigation done here that the transferability of pre-trained sequence models on other domains (vision, language), when finetuned on offline RL tasks (control, games), is novel and interesting.

**Strength And Weaknesses:**

# Strength

- The problem is well defined, with a clear mathematical formulation.
- The performance improvement is impressive.
- The paper investigates an interesting question that has a large audience.


# Weakness

- The paper fails to explain how different pretrained models influence off-line RL.
- I don't see the point of including Wikipedia in the name. I assume that it's not a key element of their model and conclusions. They may also train their model on other corpora, e.g. c4. And the paper doesn't conduct an ablation study of using a different language-based corpus.
- The interpretability of using a language model to help offline RL is weakly explained in the paper. It seems that the authors believe that the offline RL only benefits from a similar sequential structure. If so, I think of an experiment setting that may verify it. If a pretrained vision model is trained using continuous images with low resolutions, which can be constructed by dealing with videos crawled from youtube or other online channels, how will the model perform to help the off-line reinforcement learning?
- The author should provide experiments based on language models like XL-Net and RoBerta with a similar parameter size. It may help to illustrate the interpretability of using language models to help offline RL as well.
- This submission has relatively limited technical contributions, as most of the algorithmic components were proposed in previous papers.

# Questions

- The experiment results in Table. 3 are impressive but counter-intuitive. I wonder whether there is a probability that the GPU utilization is not fairly equal when running these experiments with DT, ChibiT, and GPT2. Because the experiments are done with a single V100 and models with very different parameter sizes. I hope that the authors can double-check it and provide the analysis based on steps with equal batches. The training time comparison experiment is not convincing enough for me.
- The interpretability of why the pure language-based pretraining model performs better than the pretrained vision-language multimodal is very interesting but relatively shallow.  Are there any possibilities that an experiment can be conducted to verify that the “natural” sequential nature of language and trajectories helps?

**Summary Of The Paper:**

The paper verifies that large-scale pre-training has an impressive impact on offline reinforcement learning. Accordingly, it proposes an auto-regressive based understandable language model co-training strategy ChibiT. The model has impressive performance on both Atari and GYM.

**Summary Of The Review:**

This submission is well-motivated and well-studied. The empirical results support the hypothesis that pre-trained sequence models on other domains (e.g. vision) can be transferred to other domains when fine-tuned well. Even though there are only limited technical contributions, this timely empirical investigation would be welcomed by the offline RL community.

---

> ### Author Response · Authors · 2022-11-18
> **Response**
>
> We are glad that the reviewer finds the paper interesting and well-defined, covering a novel and timely problem. We respond to their response below:
>
>
> > The paper fails to explain how different pretrained models influence off-line RL. The author should provide experiments based on language models like XL-Net and RoBerta with a similar parameter size. It may help to illustrate the interpretability of using language models to help offline RL as well.
>
> We have covered ChibiT and GPT2, and also cover how different model size affects off-line RL. The mentions of XL-Net and RoBERTa, are not comparable as RoBERTa is not autoregressive which does not fit into this current paradigm, and XL-Net is generally effective compared to a standard autoregressive model with long context, however we do not use long context in this setting (with the reasoning behind this demonstrated in our ablation).
>
> > I don't see the point of including Wikipedia in the name. I assume that it's not a key element of their model and conclusions. They may also train their model on other corpora, e.g. c4. And the paper doesn't conduct an ablation study of using a different language-based corpus.
>
> This is a valid point, we would be happy to consider suggestions by the reviewer.
>
> > The interpretability of using a language model to help offline RL is weakly explained in the paper. It seems that the authors believe that the offline RL only benefits from a similar sequential structure. If so, I think of an experiment setting that may verify it. If a pretrained vision model is trained using continuous images with low resolutions, which can be constructed by dealing with videos crawled from youtube or other online channels, how will the model perform to help the off-line reinforcement learning?
>
> Chan et al. (https://arxiv.org/abs/2205.05055), demonstrated that using a Zipfian distribution helps for learning in Transformers and we believe that this property holds with the findings of our paper. As for pre-training on YouTube, I would assume so, however the methods by which videos are represented and language is represented (as discrete tokens) is different so it is up to future research to determine how that would play a role.

---

### Official Review · Reviewer_gaVR · 2022-10-26

**Confidence:** 4
**Correctness:** 3
**Technical Novelty And Significance:** 3
**Empirical Novelty And Significance:** 2
**Recommendation:** 3

**Clarity, Quality, Novelty And Reproducibility:**

Clarity
The paper is at some points hard to follow and understand, especially Section 3.2 on the techniques where the symbols are not properly instantiated before being used.

Quality
The overall quality of the experiments is not quite up to the mark and several conlusions are drawn from very limited experimental samples.

Novelty
The idea of using pretrained language models for offline RL seems novel.

Reproducibility
I did not find the code for the method attached.

**Strength And Weaknesses:**

Strength
The overall question asked by the paper is an interesting one - whether pretraining on different modalities can be helpful for offline RL. The paper studies this problem from two perspectives: the final reward obtained by the trained policies and the computational time to achieve those rewards.

Weakness
While the idea is interesting, the paper in its current draft has several weaknesses:
- The evaluation section of the experiments is quite weak. The paper only compares the method and baselines on a Atari and a subset of the D4RL benchmarks. I would like to see a more comprehensive evalaution of the proposed method on the full suite of D4RL and RL unplugged dataset to validate the claims.
- Additionally, while the paper claims that the results are consistently better than baselines, in Table 1 for instance only for 1 (Seaquest) dataset can the proposed method be seen to perform statistically better than baselines. Bolding the means in this table is quite misleading. Similarly, for Table 2, only 3/9 cases does the method perform better.
- One of the main advantages of the proposed method is the superior convergence times as compared to exisiting benchmarks. This advantage, which is where the paper cna possibly shine, is only shown via a table on 3 environments. It would be great to actually see training curves (am not sure right now with the high variance in rewards, how the thresholding is done) across a range of environments to convince the reader that this advantage is indeed maintained.
- Section 5 (analysis section) while addresses quite a few interesting hypotheses, however, it again falls short in execution. For instance, Figure 3 is plotted for average reward across environments and the y-axis scale is from 101-104, without any error bars. I am not sure how to make a conclusive inference from this plot. Similarly, Table 4 has only two different context lengths and compares average reward (which is not an indicative measure of performance) on one particular setup. Overall, i think the all the hypotheses need to be thoroughly tested before making sweeping conlusions.
- I am surprised that setting lambda_2 to zero leads to a big degradation in performance. Does this indicate that pretraining on wikipedia is not helpful from a statistical perspective but somehow helps stabilize optimization?


**Summary Of The Paper:**

This work studies the effects of a using a pretrained language model for the offline reinforcement learning problem. It proposes techniques to improve upon the feature learning component of the pretrained models. Experimentally, this paper shows performance gains in accuracy as well as decreased training times.

**Summary Of The Review:**

The overall idea is interesting but I think the paper can be further strengthened by making the evaluations more thorough.

---

> ### Author Response · Authors · 2022-11-18
> **Response**
>
> We are glad the reviewer finds that this work tackles an important question, and understand your concerns — which we look to address in the text below:
>
> > The evaluation section of the experiments is quite weak. The paper only compares the method and baselines on a Atari and a subset of the D4RL benchmarks. I would like to see a more comprehensive evalaution of the proposed method on the full suite of D4RL and RL unplugged dataset to validate the claims.
>
> Running experiments and adapting our codebase to RL unplugged would take time that is currently not feasible given the time limit of the response period. We compare using the same set of tasks as the Decision Transformer paper, following precedent.
>
> > Additionally, while the paper claims that the results are consistently better than baselines, in Table 1 for instance only for 1 (Seaquest) dataset can the proposed method be seen to perform statistically better than baselines. Bolding the means in this table is quite misleading. Similarly, for Table 2, only 3/9 cases does the method perform better.
>
> We add performance profiles from RLiable (https://github.com/google-research/rliable) in Appendix D and would like to emphasize that the most fair comparison is with the DT, given that the objective of this paper is to investigate the impact of language pre-training on continuous control tasks. We further show that language pre-training makes DT competitive with all other methods. Following your suggestion, we have removed the boldface from Tables 1 and 2.
>
> > Section 5 (analysis section) while addresses quite a few interesting hypotheses, however, it again falls short in execution. For instance, Figure 3 is plotted for average reward across environments and the y-axis scale is from 101-104, without any error bars. I am not sure how to make a conclusive inference from this plot. Similarly, Table 4 has only two different context lengths and compares average reward (which is not an indicative measure of performance) on one particular setup. Overall, i think the all the hypotheses need to be thoroughly tested before making sweeping conlusions.
>
> The experiment with two different context lengths is meant to measure the impact of changing the context length by a relatively significant margin. However, we are open to suggestions on how to improve this experiment.
>
> > I am surprised that setting lambda_2 to zero leads to a big degradation in performance. Does this indicate that pretraining on wikipedia is not helpful from a statistical perspective but somehow helps stabilize optimization?
>
> We feel that it seems likely that pre-training helps from both a statistical perspective (Chan et al., https://arxiv.org/abs/2205.05055) and helps with stable optimization. This would be interesting to explore in more depth in future work!
>
> > Reproducibility: I did not find the code for the method attached.
>
> We have attached the code at this link https://u.pcloud.link/publink/show?code=kZV53rXZzmwveYhKpPyCLdBmShghdzYhbovV

---

> > ### Comment · Reviewer_gaVR · 2022-12-08
> > **post response update**
> >
> > Thanks a lot for responding back to the questions in the reviews. Thanks for providing a link to the code for reproducibility!
> >
> > After going through the other reviews and responses, I still believe that the paper has a very interesting and surprising idea but lacks a bit in execution --
> > 1) The ablation studies performed in Section 5 can be expanded to a larger set of datasets. Stating the hypothesis that each ablation is trying to check and running a statistical test to validate that (on multiple datasets) would provide more information on the ablation.
> > 2) I think a preliminary analysis (and comparison) of the training curves for the proposed method with decision transformer should possibly help us figure out how it helps optimization.
> > 3) From the statistical perspective, it would be good to include plots indicating performance curves with the amount of offline data available, comparing it with baseline approaches to see whether the proposed method performs better in say a low sample or high sample regime.

---

### Official Review · Reviewer_F11T · 2022-11-01

**Confidence:** 4
**Correctness:** 4
**Technical Novelty And Significance:** 3
**Empirical Novelty And Significance:** 3
**Recommendation:** 6

**Clarity, Quality, Novelty And Reproducibility:**

The article is of high quality, clearly written and original, this time exploring for the first time the novelty of transferring a pre-trained generative model from one domain (language) to another completely different domain.

**Strength And Weaknesses:**

Strengths
1. The pre-trained model is built under the Decision Transformer framework, which is a very novel attempt, and the scheme is feasible from the experimental results.
2. The experimental parameters are comprehensive, reproducible and technically very solid, which shows that the authors have sufficient technical reserves and good insights into the field, and the information provided represents a fair effort to enable the reader to reproduce the results.
3. The authors demonstrate the excellent performance of the two proposed pre-training models through comprehensive experiments, revealing the potential of pre-training models using generic sequence modeling techniques and RL, inspiring future work on knowledge sharing between generative modeling tasks in completely different domains.

Weaknesses
1. too little is shown about the model architecture, can a more specific discussion be given?
2. the convergence speed of the model proposed in this paper is significantly reduced relative to the Decision Transformer, which proves the effectiveness of pre-training but does not consider the time required for pre-training, can this be provided?
3. In terms of the final results, the reward boost is not groundbreaking, can you explain why this is the case? Is it due to data quality or model architecture?

**Summary Of The Paper:**

This paper proposes a technique to improve inter-domain migration after investigating how pre-trained models can improve the general offline RL problem, using reinforcement learning as a paradigm for sequence modeling, and investigating the transferability of pre-trained sequence models on other domains (vision, language) for fine-tuning on offline reinforcement learning tasks (control, games). The authors' fine-tuning from a Wikipedia-trained small transformer (ChibiT) and GPT2 model proves to be substantially better than the basic Decision Transformer (DT) and other RL-based offline baselines in terms of policy performance and convergence, building state-of-the-art scores on competitive D4RL benchmarks from Gym and Atari, and reducing DT training time by a factor of 3-6, clearly demonstrating that language pre-training outperforms random initialization using sequence modeling techniques in terms of reward. The work is somewhat innovative, transferring a pre-trained generative model in one domain (language) to a generative modeling task in another completely different domain (RL on continuous control and games), which is an effective attempt in terms of experimental results. This is a recommended paper that reveals the potential of pre-trained models using generic sequence modeling techniques and RL.

**Summary Of The Review:**

This paper address offline reinforcement learning from a sequence modeling perspective with an eye to introducing the Transformer architecture, and the results are improved to address the problem of high variability in migrability between different environments. When models are trained from scratch, convergence is slow, and the authors use this reinforcement learning as sequence modeling to investigate the transferability of pre-trained sequence models on other domains (vision, language) for fine-tuning offline reinforcement learning tasks (control, games). I have some knowledge of the relevant techniques and the overall architecture is reasonable.

---

> ### Author Response · Authors · 2022-11-18
> **Response**
>
> We are happy that the paper appreciates the potential of using pre-trained sequence models for RL, and finds the experiments to be comprehensive and technically solid. We respond to concerns below.
>
>
> > Too little is shown about the model architecture, can a more specific discussion be given?
>
> As stated in Section 4.1, we state that our model is a standard decoder transformer similar to that used in the Decision Transformer and original Transformer papers. If there is anything more specific still unclear, please let us know.
>
>
> > The convergence speed of the model proposed in this paper is significantly reduced relative to the Decision Transformer, which proves the effectiveness of pre-training but does not consider the time required for pre-training, can this be provided?
>
> Yes, we provide this in our hyperparameter section and footnote 5, in which we train for a maximum of 6 hours on a single GPU. Note however, that this single backbone can be reused for multiple environments and settings without pre-training again.
>
> > In terms of the final results, the reward boost is not groundbreaking, can you explain why this is the case? Is it due to data quality or model architecture?
>
> We believe that this is largely due to the disparity in data modality. We believe that if more data matched the modality of our downstream tasks, then performance boosts would be greater. However, this paper serves to make a point that pre-training on language does indeed provide useful benefits.

---

### Author Response · Authors · 2022-11-18
**Response**

We thank all the reviewers for their insightful comments and suggestions. We appreciate that the reviewers find the experimental evaluation to be comprehensive and solid (RF11T, Ry7eL, R3DmF), while finding the idea to be interesting (RgaVR, RF11T, Ry7eL, R3DmF). We look to address concerns in the individual comments and thank the reviewers for making the paper stronger in the process.

---

### Decision · Program_Chairs · 2023-01-20

**Decision:**

Reject

**Justification For Why Not Higher Score:**

Lack of rigorous evaluation to justify the claims in the paper.

**Justification For Why Not Lower Score:**

N/A

**Metareview: Summary, Strengths And Weaknesses:**

The paper investigates the benefits of cross-modal pretraining for offline RL. Specifically, the paper finds that a GPT model pretrained on Wikipedia can generalizes better and faster on the D4RL environments than a related baseline (Decision Transformers). On the positive side, there was appreciation of the topic of the study extending work from Lu et al. (2021) to the offline RL context and the empirical gains observed over DT. However, post rebuttal and discussions, there were also outstanding reviewer concerns on the generality of the method and the statistical rigor of the claims. The primary comparison is made against DT, but the performance gains in Table 1 are small and in fact, compared against a somewhat outdated codebase. Many training tricks have been proposed to improve DT (e.g., https://arxiv.org/abs/2112.10751, https://arxiv.org/abs/2202.05607 -- see offline only numbers in Table 5.1) which show numbers competitive with Chibit/GPT2, making it unclear how significant are the proposed benefits. With regards to improved convergence, there were many reviewer concerns on the rigor of this work: the lack of training curves and error bars to understand the improved convergence, the adhoc learning tricks (eg, decay coefficients to 0) that were not empirically ablated, and the absence of evaluation of these convergence claims beyond 3 D4RL datasets (e.g., on Atari, other D4RL datasets). While there is potentially merit in the work, it is difficult to accept it in the current form due to the lack of rigorous evaluations, which is especially critical for RL. I encourage the authors to address these issues for a future submission.